# Considering Alternate Pathways of Drinking-Water Contamination: Evidence of Risk Substitution from Arsenic Mitigation Programs in Rural Bangladesh

**DOI:** 10.3390/ijerph17155372

**Published:** 2020-07-26

**Authors:** Varun Goel, Griffin J. Bell, Sumati Sridhar, Md. Sirajul Islam, Md. Yunus, Md. Taslim Ali, Md. Alfazal Khan, Md. Nurul Alam, ASG Faruque, Md. Masnoon Kabir, Shahabuddin Babu, Katerina Brandt, Victoria Shelus, Mark D. Sobsey, Michael Emch

**Affiliations:** 1Department of Geography, University of North Carolina-Chapel Hill, Chapel Hill, NC 27514, USA; kebrandt@live.unc.edu; 2Gillings School of Global Public Health, University of North Carolina-Chapel Hill, Chapel Hill, NC 27599, USA; gjbell86@live.unc.edu (G.J.B.); vshelus@live.unc.edu (V.S.); mark_sobsey@unc.edu (M.D.S.); 3Department of Statistics, University of North Carolina-Chapel Hill, Chapel Hill, NC 27514, USA; ssridhar000@gmail.com; 4International Centre for Diarrhoeal Disease Research (icddr,b), Dhaka 1212, Bangladesh; sislam@icddrb.org (M.S.I.); myunus@icddrb.org (M.Y.); taslim@icddrb.org (M.T.A.); fazal@icddrb.org (M.A.K.); nalam@icddrb.org (M.N.A.); gfaruque@icddrb.org (A.F.); tooha.btech@gmail.com (M.M.K.); shahab.babu@gmail.com (S.B.)

**Keywords:** drinking water quality, Bangladesh, household water storage, arsenic mitigation, risk substitution

## Abstract

Deep tubewells are a key component of arsenic mitigation programs in rural Bangladesh. Compared to widely prevalent shallow tubewells, deep tubewells reduce ground-water arsenic exposure and provide better microbial water quality at source. However, the benefits of clean drinking-water at these more distant sources may be abated by higher levels of microbial contamination at point-of-use. One such potential pathway is the use of contaminated surface water for washing drinking-water storage containers. The aim of this study is to compare the prevalence of surface water use for washing drinking-water storage containers among deep and shallow tubewell users in a cohort of 499 rural residents in Matlab, Bangladesh. We employ a multi-level logistic regression model to measure the effect of tubewell type and ownership status on the odds of washing storage containers with surface water. Results show that deep tubewell users who do not own their drinking-water tubewell, have 6.53 times the odds [95% CI: 3.56, 12.00] of using surface water for cleaning storage containers compared to shallow tubewell users, who own their drinking-water source. Even deep tubewell users who own a private well within walking distance have 2.53 [95% CI: 1.36, 4.71] times the odds of using surface water compared to their shallow tubewell counterparts. These results highlight the need for interventions to limit risk substitution, particularly the increased use of contaminated surface water when access to drinking water is reduced. Increasing ownership of and proximity to deep tubewells, although crucial, is insufficient to achieve equity in safe drinking-water access across rural Bangladesh.

## 1. Introduction

Most rural Bangladeshi residents consume drinking-water from groundwater aquifers from tubewells. Of rural households, 74% consume water from shallow tubewells with depths less than 46 m [1]. Shallow tubewells are relatively inexpensive, privately owned and conveniently located; they generally provide access to clean drinking-water compared to surface water [2]. However, many of these tubewells exposed rural residents to high levels of naturally occurring arsenic, which led households to switch from contaminated shallow tubewells to nearby low arsenic shallow tubewells to mitigate arsenic exposure [3,4]. Follow-up studies showed that such shallow tubewells were more likely to be contaminated with fecal organisms due to the nature of local hydrogeology and factors such as high population density, and a higher risk of contamination due to poor neighborhood sanitation and hygiene conditions [5,6,7,8]. To avoid such concerns, some residents switched to consuming drinking-water from deep tubewells greater than 152 m deep that tap into aquifers mostly free of arsenic [9]. As per the 2014 Bangladesh Demographic Health Survey, one out of four rural households in Bangladesh consumes drinking-water from a deep tubewell [1], with over 165,000 installed throughout the country [10]. Deep tubewells have also demonstrated lower levels of microbial contamination at source due to increased depth [5,8], and have been associated with a lower incidence of diarrheal diseases compared to shallow tubewells [11,12].

There is growing evidence that although deep tubewells are cleaner at source compared to shallow tubewells, access to them does not guarantee household consumption of safer water. Water storage and household water management practices can compromise water safety and increase point-of-use (POU) microbial contamination [13,14,15,16]. Our previous work from an ongoing study found increased POU microbial contamination among deep tubewell users compared to shallow tubewell users [17]. Because many of these deep tubewells are more expensive and thus publicly installed, they tend to be installed near a road or in a central location. These tubewells are often inequitably distributed; some villages have multiple deep tubewells while others do not have any [18]. Previous studies found that increasing ownership and density of deep tubewells in high arsenic areas can present a viable cost-effective option to provide safer drinking water to residents of rural Bangladesh [19], and have significant positive health and economic impacts [20].

In summary, previous research indicates that increased deep tubewell access and density may provide better domestic household-level microbial quality. However, in addition to the ‘domestic’ household level domain such as household water storage and handling, the ‘public’ domain such as filling and washing of storage containers from a polluted water source can also affect POU microbial contamination [21]. Empirical gaps remain in calculating risks due to alternate pathways, such as the washing of drinking-storage containers, that lie at the intersection of domestic and public domains of microbial contamination. One study in Punjab, Pakistan, reported that contamination attributed originally to household storage was associated with the washing of water storage containers [22]. Those findings may have relevance for rural Bangladesh, where households typically use tubewell water for drinking purposes and surface water for some non-drinking purposes [23]. Such additional risk through the ‘public’ domain may be re-introduced by washing drinking-water storage containers with surface water because such water is often highly microbially contaminated [24,25,26,27]. Hence, interventions such as increasing tubewell access to ensure safer microbial quality need to address both domestic and public pathways, especially in the case of washing drinking-water storage containers, where these pathways intersect. 

To our knowledge, this is the first study to examine differences in exposure among deep tubewell and shallow tubewell users to potentially riskier practices of washing drinking-water storage containers with surface water. We also assess the effect of reduced access to deep tubewells on the likelihood of washing drinking-water storage containers compared to other users differing on tubewell type and ownership status. Our study aims to fulfill an important empirical gap and highlight other potential dimensions of risk among deep tubewell users and provide suggestions for interventions to address resulting policy gaps. 

The primary objectives are to (1) compare the prevalence of surface water use for washing drinking-water storage containers among deep tubewell users compared to shallow tubewell users and (2) determine the effect of tubewell type and tubewell ownership of drinking water source on using surface water for washing drinking-water storage containers.

## 2. Materials and Methods 

### 2.1. Study Design

We conducted a cohort study of rural households using shallow tubewells or deep tubewells as their primary drinking-water source. Using a stratified random sampling design, we assigned households to two groups, deep tubewell users and shallow tubewell users. For each household, we measure the microbial quality of the drinking-water using samples collected from the drinking-water source and the POU container to compare microbial water quality among the two groups, both during the rainy season and the dry season. We supplement microbial quality with surveys to assess diarrhea prevalence among children under-five years of age and measure the distance between the tubewell and the household using global positioning system (GPS) units [17]. In this paper, we consider a cross-section of the study and utilize the baseline survey during the rainy season to measure the prevalence of surface water use to wash storage containers among deep tubewell users and shallow tubewell users.

### 2.2. Setting

The study area is in Matlab, Bangladesh, a rural area in southeastern Bangladesh, located 50 km southeast of Dhaka, which comprises 142 villages and approximately 56,000 households. Based on the last large-scale arsenic testing campaign in 2002–03, Matlab was characterized by high concentrations of arsenic in shallow aquifers. Out of the 12,500 tubewells tested for arsenic across the study area, the majority being shallow tubewells with depths between 0–140 feet, 64% of tubewells contained arsenic in concentrations above the Bangladeshi permissible limit of 50 μg/L [28,29]. The main arsenic mitigation strategy for households in the study area has been to switch to neighboring low arsenic shallow tubewells, followed by switching to deep tubewells, wherever available. According to the 2014 Matlab Household Socio-economic Census, 84% of households reported procuring drinking-water from a shallow tubewell, compared to 15% of households that reported using a deep tubewell [30]. Although there has been no major arsenic testing campaign since, 32% of households reported using a shallow tubewell that was either untested or marked as unsafe during the 2002–03 campaign [30].

For our study, we identified households with under-five children using the Matlab Health and Demographic Surveillance System (HDSS) [31] and a follow-up survey. Households were recruited and surveyed at baseline by local community health workers during the second half of the rainy season between July and September 2016. Follow-up surveys regarding water quality and tubewell use were implemented once during the rainy and dry seasons through July 2019. We use the data from the baseline during the rainy season for the current analysis. All subjects gave their informed consent for inclusion before they participated in the study. The study was conducted in accordance with the Declaration of Helsinki, and the protocol was approved by both the Institutional Review Board of the University of North Carolina-Chapel Hill (Study No. 16-0298) and the Ethics Research Committee of the International Centre for Diarrhoeal Disease Research, Bangladesh (ICDDR,B) (PR-16081).

### 2.3. Eligibility Criteria

We initially selected households who identified either a shallow or a deep tubewell as their primary drinking-water source in the 2014 Matlab Socio-economic Census and contained at least one child born on or after 31 July 2016. Follow-up interviews for those households were conducted during the baseline survey and those not satisfying the eligibility criteria were excluded. In case of absence, migration, or exclusion, we interviewed a neighboring household in the same *bari*. *Baris* are clusters of households that are arranged spatially based on patrilineal linkages. Generally, members in the same bari drink from a common tubewell. Baris that do not own a tubewell usually collect drinking-water from wells owned by a neighboring bari or a community well such as those at a mosque or school.

### 2.4. Variables and Measurement

The outcome for the analysis is a binary variable that identifies whether a household self-reports using surface water to clean their main drinking-water storage container. Households reporting mixing of water sources such as surface water and shallow tubewell water were also quantified as users of surface water. The type of tubewell used was the exposure variable. Although it is possible that households may intermittently use different tubewell sources for drinking, most households reported primarily using one tubewell for their drinking purposes throughout. Subsequently, the tubewell type was based on the primary well from which households usually fetch their water. In the statistical analysis, we considered the ownership status of the primary drinking-water tubewell as a potential effect modifier. Ownership status indicated whether the household owned the tubewell they reported using for consumption in their bari (referred to as inside bari) or not (referred to as outside bari). Households that acquired water from either another bari or a community tubewell were considered to not own their primary drinking-water source. We expected tubewell ownership to be correlated with distance from the household to the tubewell of interest. Furthermore, tubewell ownership conveys additional information, which distance does not. For example, households that do not own a tubewell might use a tubewell less frequently because of potential social barriers to using a tubewell owned by another household. For the more expensive deep tubewells, households might also not be able to contribute to the cost if the tubewell goes into disrepair. For these reasons, we chose to model ownership, instead of distance, in the analysis. We also included the socio-economic status (SES, in quintiles) for the households and the number of years of the mother’s education as control variables. Both variables were obtained from the HDSS from 2014.

### 2.5. Statistical Methods

The analysis for the study was conducted in R 3.6.1 using the lme4 package version 1.1-21 [32,33]. We measured the odds ratios of the probability of surface water use for washing storage containers based on tubewell type before and after adjusting for covariates. A multi-level logistic model was fit to the data from households with no missing data (*n* = 488), with the odds of surface water use, as previously defined, as the outcome of interest. Model covariates included tubewell type, tubewell ownership, household SES, mother’s education, and the interaction between tubewell type and tubewell ownership. A random intercept for each village was included to account for correlation among households from the same village. We also ran a point biserial correlation test between ownership and distance, to formally test for multi-collinearity due to distance and ownership.

## 3. Results

### 3.1. Descriptive Statistics

We included 499 households in the analysis, belonging to 485 unique baris and 118 unique villages. Out of the 499 households, 253 used shallow tubewells as their primary drinking-water source and 246 used deep tubewells. Table 1 displays descriptive information for the households in this sample. Of the shallow tubewell users, 28.45% used surface water to wash their drinking-water storage container compared to 50.41% of deep tubewell users. Most shallow tubewell users (83.79%) owned the tubewell that they reported drinking from, while less than half (43.90%) of deep tubewell users did. The distribution of SES levels and the mother’s education levels were similar for both shallow and deep tubewell users. 11 (8 deep, 3 shallow) participants had missing data for SES and education level. The correlation coefficient between ownership and distance was 0.615 (*p* value < 0.0001) suggesting that ownership and distance are highly correlated.

### 3.2. Model Results

We estimated both crude and adjusted odds ratios of surface water use to clean drinking-water containers to compare four categories based on tubewell and ownership type of shallow or deep tubewells. These odds ratios are displayed in Table 2.

Adjusted odds ratios remain similar after adjusting for SES and the mother’s education. Conditional on owning a tubewell, deep tubewell users had 2.53 (1.36, 4.71) times the odds of using surface water to wash drinking-water storage containers than shallow tubewell users. Conditional on not owning a tubewell, deep tubewell users were less likely to use surface water, but with odds ratios ranging from 0.30, suggesting lower likelihood, to 1.63, suggesting higher likelihood, being reasonably compatible with the data, given the model assumptions. Among shallow tubewell users, those who did not own a tubewell had 9.42 (4.04, 21.96) times the odds of using surface water compared to those who owned a tubewell. Among deep tubewell users, those who did not own a tubewell were more likely to use surface water, with an odds ratio of 2.58 (1.40, 4.74). Deep tubewell users who did not own their tubewell were more likely to use surface water than shallow tubewell users who owned their tubewells, with an odds ratio of 6.53 (3.56, 12.00). Shallow tubewell users who did not own their tubewell were more likely to use surface water than deep tubewell users who owned their tubewells, with an odds ratio of 3.72 (1.53, 9.03). 

## 4. Discussion

### 4.1. Summary of Results

We found that half of the deep tubewell users clean their drinking-water storage containers with surface water compared to less than one-third of shallow tubewell users. Hence, in addition to the domestic pathways of microbial contamination such as longer drinking-water storage times in containers, public pathways such as contamination from an external non-drinking source (surface water) may also be responsible for increased microbial contamination observed in our previous study [17]. Cross-contamination in the storage container could occur due to the ‘mixing’ of multiple water sources [34,35] (surface water and tubewell water) or due to improper rinsing of containers while cleaning [36].

The SES of the household and the level of the mother’s education were not significantly different across deep and shallow tubewell users, suggesting those variables do not influence whether a household drinks water from a shallow or a deep tubewell. The ownership status of a tubewell, however, is an extremely important and influential factor in determining the odds of whether deep or shallow tubewell household members wash their container with surface water. Both shallow and deep tubewell users that do not own their primary drinking-water source are at higher odds compared to their counterparts that report owning their drinking water source. These results are consistent with findings from other areas of rural Bangladesh that show that well-sharing as an arsenic mitigation strategy may not be a widely accepted option [37], and increase barriers to access for households that do not own a tubewell that is safe for drinking. Strikingly, the majority of deep tubewell users, who also happen to procure drinking-water outside their baris, are at even higher odds of using surface water to wash storage containers, when compared to the majority of shallow tubewell users, who own their tubewells.

These findings are significant because, despite the increased physical distance to their drinking water source (77.2 m median distance for non-owner deep tubewell users compared to 12.2 m for shallow tubewell users, who own their wells), most households also own or have a shallow tubewell nearby to use for non-consumption purposes. Hence, although we expected lower use of deep tubewells for non-consumption purposes, we also expected general tubewell use for non-consumption purposes such as washing storage containers to be similar among households using deep or shallow tubewells for drinking purposes. Although rural Bangladeshis may prefer surface water for non-consumption purposes compared to tubewells due to habit, traditional norms, and staining due to groundwater use [23], these differences do not explain the difference in surface water use among the households. One possible explanation based on correspondence with local community health workers who conducted the survey is that since deep tubewell users travel farther and take fewer trips and use larger containers to store their drinking water [17], it may be easier to wash those storage containers in surface water by completely submerging them rather than cleaning them under the mouth of a tubewell. Another possible explanation could be that non-owner deep tubewell users, who have presumably switched from shallow tubewells due to arsenic contamination, may also be more likely to use surface water for washing storage containers and other non-consumption purposes due to concerns of further arsenic contamination and ingestion. Future research is needed to address disparities in surface water use for washing-storage containers among deep tubewell and shallow tubewell users.

We also find that households that drink water from deep tubewells inside their bari are still at higher odds of washing their drinking-water container with surface water compared to shallow tubewell owners. Physical distance does not explain these differences since there is not a significant difference in median distance to the tubewell for deep tubewell users (17.7 m) compared to shallow tubewell users (12.2 m). Hence, it is possible that despite the advantages of deep tubewells for drinking purposes, households may perceive deep tubewells to have certain disadvantages for non-drinking purposes. In our sample, some deep tubewell users reported more effort and time required to pump drinking-water from the deep aquifer. Additionally, in some parts of the study area, respondents noted a red color in deep tubewell water due to the presence of iron, which may deter them from using deep tubewell water for washing purposes. Although we did not test for such factors, rural Bangladeshis may have different preferences for water used for consumption and non-consumption purposes [23]. Odor, color, and effort required to pump from deep tubewells may all play a role in households restricting their deep tubewell use only for drinking and not for other purposes such as washing their storage containers.

### 4.2. Significance and Policy Implications

The findings add an important novel dimension to the evidence related to the idea of risk substitution, that arsenic mitigation programs in Bangladesh and other affected countries may result in increases in health risks due to waterborne pathogens. So far, studies have shown that switching to lower arsenic deep tubewells is associated with both an increase [38] and a decrease [11,12,39] in health risks associated with diarrheal diseases. Even results from the large multi-site *Global Enteric Multicenter Study* (*GEMS*), conducted in two separate sites close to our study area, have been variable; deep tubewells were associated with a higher incidence of less severe *Shigella sonnei* infections compared to shallow tubewell users in one study site [40], while there was no difference in diarrheal diseases stratified by tubewell type in another [41]. However, none of these or other studies have yet investigated the mechanisms through which deep or shallow tubewell users may have differential health risks. This is important since it is possible that despite similar diarrheal disease outcomes, the pathways associated with diarrheal disease risk among deep and shallow tubewell users may be different. In a previously published paper, we highlighted the role of physical distance and higher storage times among deep tubewell users with increased microbial contamination in drinking-water storage containers [17]. This analysis highlights another critical pathway of microbial contamination risk: the increased use of surface water for washing storage containers among deep tubewell users. Washing storage containers with highly contaminated surface water can help explain the introduction of *E. coli* at POU in households that drink from tubewells free from contamination at the source. Importantly, our analysis also highlights the role of social factors such as ownership of a tubewell and the need to study other social, political, and behavioral determinants in addition to physical distance barriers that increase exposure among deep tubewell users.

From a policy perspective, our study provides new information and insights into the role of several key factors associated with household water collection and use for tubewell water users, specifically tubewell type (shallow or deep), tubewell ownership and extent of hygienic management of drinking-water collection storage containers, specifically use of surface water to clean the container. Increasing deep tubewell density and reducing physical distance, especially in areas with high arsenic content in the shallow aquifer, will greatly reduce the odds of washing storage containers with surface water. In addition to physical distance, ownership considerations too, are important. An increase in privately installed deep tubewells has in part been driven by *elite capture* where households with political influence are able to divert public funds to install a tubewell within their baris [18]. Although it is a common norm among rural Bangladeshis to freely consume groundwater from the tubewells installed with public funds, the inequitable ownership of these wells may deter households from using deep tubewells for purposes other than drinking and being asked to financially contribute to tubewell maintenance. Hence, the continuous involvement of governmental and non-governmental agencies is required to protect vulnerable households and maintain equity in the provision of safe drinking water. Other ways to address water safety need to be considered too since according to our results, increased ownership of deep tubewells may not be adequate; chances of contamination through the public transmission route from surface water may remain significantly high. Policies and practices for drinking water, including the collection and storage of household water are best addressed through the Water Safety Plans (WSPs) of the World Health Organization (WHO) that address drinking-water quality and safety based on comprehensive user-friendly guidance. For example, although household level arsenic treatment measures are widely accepted, more than 80% of rural Bangladeshi do not treat their drinking water for other microbes [42]. WSPs for collected and stored water in households are limited in use and many countries lack user-friendly guidance in ways that reach and are actionable for user communities and households. The development of such WSPs with the help of WHO in countries where such guidance does not exist or is inaccessible is necessary, and this study can inform the development and implementation of such guidance.

Despite moderate availability and use, deep tubewells are a widely supported option for mitigating arsenic and providing safe drinking water. When surveyed, current and potential deep tubewell users, including shallow tubewell users, rate deep tubewells highly for psychological and socio-cultural reasons, and institutional stakeholders support them due to their stable water quality and source [19,37,42,43,44]. Our proposed policy implications and suggestions are consistent with those findings and are especially relevant in southern Bangladesh, a region with high arsenic contamination in shallow aquifers, and low groundwater depth [45,46]. Multiple studies show that deep tubewells are a sustainable source of arsenic mitigation, and as long as groundwater use from deep wells is restricted to domestic, as opposed to agricultural use, deep aquifers could provide safe arsenic-free water for many decades [47,48,49]. Although more than 80% of the water used for dry-season irrigation is pumped from shallow tubewells [50,51], the use of deep tubewells for agricultural supply has increased over time, primarily due to health concerns of arsenic in rice production [52,53]. Groundwater flow modeling suggests that deep irrigation pumping could induce arsenic into deep aquifers from high arsenic shallow aquifers through a downward flow [54]. Although such concerns are important for the long term sustainability of deep tubewells to provide safe drinking-water, Ravenscroft et al. [55] emphasize the need to balance short-term unsustainable extraction with the adverse multi-generational public health impacts of arsenic toxicity and water-borne diseases among rural Bangladeshis. Hence, although the feasibility of deep tubewells in the present and future depends on a combination of geological and health factors, the findings in this paper are generalizable to any setting where deep tubewells are currently or potentially going to be used as an arsenic mitigation option. These settings not only include rural Bangladesh, and neighboring areas with established groundwater arsenic issues such as eastern India and Myanmar, but also other places such Afghanistan, Laos, Pakistan, and Thailand [56].

### 4.3. Limitations and Future Work

This study has some important limitations. First, the analysis is currently based on cross-sectional data collected in the rainy season and does not currently take seasonality into account. It is possible that use of surface water for washing storage containers and its relationship to tubewell use may differ in the dry season, especially since surface water availability during the dry season is limited and may prompt households using deep tubewells previously using surface water to use tubewell water to wash their storage containers. Second, since we were interested in the total effect of ownership and tubewell type, we did not include mediator variables that could help parse out the specific pathways through which ownership and tubewell type influence health risks. In our future research in this ongoing cohort study, we will conduct a mediation analysis on the longitudinal data to parse out the multiple domestic and public causal pathways of contamination. This would help us untangle composite variables such as ownership, and help assess the most salient pathways through which health and microbial contamination risk are propagated. For example, by using mediation analysis we will formally be able to test under what conditions physical barriers such as long walking distance to a deep tubewell may be more important than social-cultural barriers such as lack of control over tubewell use and informal norms. Finally, although we were not able to analyze how surface water use translates to water quality and subsequent health outcomes in this paper, future research will include a longitudinal analysis to examine how different pathways are associated with water quality and diarrheal disease risk. 

## 5. Conclusions

Our findings suggest that the potential for cross-contamination due to the cleaning of drinking-water storage containers with surface water among deep tubewell users in rural Bangladesh is substantial and needs to be considered while evaluating the health impacts of deep tubewells as part of arsenic mitigation programs. In terms of interventions to ensure safe drinking water consumption from deep tubewells, external public pathways of contamination from external sources such as surface water should be addressed in addition to domestic pathways of safe water storage and handling. We argue, however, that while separate interventions targeting these two separate pathways (domestic and public) are important [21,22], addressing distal factors such as barriers to access (distance and ownership) has the potential to simultaneously address both pathways and reduce both longer storage times and surface water use in cleaning storage containers. There is an important need to consider ownership of tubewells, in addition to distance in future studies to fully address barriers to access and its impact on health. Studies from Bangladesh and beyond, have highlighted different methods to promote safe household drinking water storage. They include the need to target the neighborhood hygiene and sanitation environment, promote education and behavioral interventions, or target separate interventions for direct and indirect pathways to reduce microbial contamination and diarrheal disease risk [57]. In addition to these interventions, policy makers and researchers should consider the complex role and potential for substantive trickle-down effects of providing increased and equitable access to deep tubewells and ensure safe drinking water quality for all. Additionally, there is a need for the development and implementation of comprehensive, but user friendly WSPs and accessible supporting tools as practical guidance for safe water management in such communities and households.

## Figures and Tables

**Table 1 ijerph-17-05372-t001:** Summary of Households by Tubewell Type.

Variables	Shallow (*n* = 253)	Deep (*n* = 246)
Households using surface water for washing storage containers. (%)	72 (28.45)	124 (50.41)
**Tubewell Ownership (%)**		
Inside Bari	212 (83.79)	108 (43.90)
Outside Bari	41 (16.21)	138 (56.10)
**SES (%)**		
(Poorest) Quantile 1	39 (15.41)	37 (15.04)
Quantile 2	45 (17.79)	47 (19.11)
Quantile 3	44 (17.39)	45 (18.29)
Quantile 4	66 (26.09)	55 (22.36)
(Richest) Quantile 5	56 (22.13)	54 (21.95)
Missing	3 (1.19)	8 (3.25)
**Mother’s Education in Years**		
Mean (SD)	3.91 (3.84)	3.94 (3.96)
Missing (%)	3 (1.19)	8 (3.25)

**Table 2 ijerph-17-05372-t002:** Odds Ratios (OR) comparing the use of surface water while washing drinking-water storage containers by tubewell and ownership type.

Comparison Group	Reference Group	Crude OR (95% CI)	Adjusted OR (95% CI)
Deep Tubewell (Inside Bari)	Shallow Tubewell (Inside Bari)	2.21 (1.21, 4.04)	2.53 (1.36, 4.71)
Deep Tubewell (Outside Bari)	Shallow Tubewell (Outside Bari)	0.70 (0.31, 1.62)	0.69 (0.30, 1.63)
Shallow Tubewell (Outside Bari)	Shallow Tubewell (Inside Bari)	10.53 (4.57, 24.24)	9.42 (4.04, 21.96)
Deep Tubewell (Outside Bari)	Deep Tubewell (Inside Bari)	3.35 (1.89, 5.95)	2.58 (1.40, 4.74)
Deep Tubewell (Outside Bari)	Shallow Tubewell (Inside Bari)	7.40 (4.07, 13.45)	6.53 (3.56, 12.00)
Shallow Tubewell (Outside Bari)	Deep Tubewell (Inside Bari)	4.77 (2.01, 11.30)	3.72 (1.53, 9.03)

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
