# Peer review of "Considering Alternate Pathways of Drinking-Water Contamination: Evidence of Risk Substitution from Arsenic Mitigation Programs in Rural Bangladesh"

_ijerph, 2020, doi:10.3390/ijerph17155372_

Round 1

Reviewer 1 Report

Overall, the manuscript is well-conceived, nicely organized, conveys main points mostly clearly, and has the potential to make an important contribution to better understanding tradeoffs in risks (i.e. arsenic/microbes) associated with drinking water and its conveyance to households in rural Bangladesh.

The authors might consider including additional background information including what is known about the available groundwater resources at different depths in the study area; some aquifers may already be drying up. It is very unlikely that a significant increase in use of deeper aquifers would be sustainable in the long run as recharge rates are generally much slower than those of surficial aquifers.  It is unclear from reading this manuscript whether the deeper aquifers are only being used for drinking water purposes or also for other (e.g. agricultural) uses, which may deplete the “safer” aquifers more quickly?

There are some instances in the manuscript that could be worded more clearly—for example (there are others):

Page 1, line 20: I don’t believe there is such a thing as a source of a mitigation program—there are sources of contamination, however. I believe the authors mean that deep tubewells are a key component of arsenic mitigation programs?

Page 1, line 30 and line 32— it would be more understandable to substitute the phrase “more likely than” for “the odds compared to”

Page 1, line 35: Increasing ownership of and decreasing distance to…?

Page 3, line 122: a household cannot consist of at least one child—a household could contain at least one child…or we selected households in which at least one child was present?

The main shortcoming of this manuscript is that the authors mainly focus on tubewell ownership despite (as stated in section 2.1) having data on distance between households and drinking water sources.  I am in no way arguing that ownership is unimportant. But the authors later state that individuals/households are likely making fewer trips with larger containers (and likely washing with surface water in-between) to non-owned wells.  The distance data could help understand whether this is the case—It is likely that deep tubewell distance from households varies significantly and incorporating this into the analysis may help tease out whether ownership or distance is more important.  In other words, is the physical distance more important than the social distance (i.e. not owning the well or the well is owned by others outside the familial or social group)?  If distance (or ownership) ends up being the more powerful variable, this might help inform efforts to continue reducing arsenic exposure while also reducing microbial contamination and diarrhea prevalence.  It is possible that the authors have done this analysis already and did not include it here or included it elsewhere (another publication) but if so, it would be nice to be able to discuss it in this manuscript as well.

Reviewer 2 Report

The article is interesting from the point of view of water safety. However it cancels basic information about:

-the quality of water from shallow and deep wells, what is the concentration of arsenic, iron, manganese, microbial cotamination in both type of water sources,

-surface water quality and availability  if it,

-the analyzes of water from storage containers are not presented,

-statistics of cases of water born diseases,
-water prices.

Supplementing the article with the above data would increase the scientific level of the article and allow to base conclusions also on the results of chemical and microbiological water quality analyzes.

Author Response

Response to Reviewer 2 (R2)

We thank R2 for helping us think about strengthening our results by adding evidence from chemical and microbial data and addressing limitations.

  1. Supplementing the article with data on chemical and microbiological water quality for both tubewells, availability and quality of surface water, analysis of water from storage containers, statistics of water-borne diseases and water prices. Base conclusions on the results of chemical and microbiological water quality analysis.

We thank R2 for providing suggestions on strengthening the conclusions based on chemical and microbiological water quality analysis. We agree with R2’s suggestions but cannot fully address them due to limitations of our study. We would also like to state that while a more comprehensive analysis is needed to both tease out the casual pathways of risk substitution, their impact on microbial water quality, and the relationship on diarrheal disease, the focus of this paper is to highlight one particular exposure pathway: the use of surface water for washing drinking-water storage containers. Thus, we restrict the analysis to modeling exposure to the use of surface water among shallow and deep tubewell users. However, we have substantially expanded our discussion section to highlight how our findings are related to available evidence on chemical and microbiological water quality. Our previous study by (Goel et al., 2019), had found that deep tubewell users had higher microbial contamination in drinking-water storage containers compared to shallow tubewell users, despite comparable water quality at source. In that study, we analyze storage times and walking distance as important factors linked to microbial quality. We synthesize those results with our current findings and show that washing storage containers with highly contaminated surface water can help explain the introduction of E. coli at point-of-use in households that drink from tubewells free from contamination at source (section 4.2. lines 282-290). Although we did not measure surface water as part of our tubewell study, we conducted a pilot to compare surface water quality with tubewell water quality and found that almost all surface water samples were contaminated. In this paper we have added several references from similar studies in Bangladesh to show that surface water is often highly contaminated (Section 1 – line 84). We believe that these additions help justify the importance of our findings without the need to add more data in our current analysis. However, we have re-written our limitations and future work section (4.3 lines – 373 to 391), and note our future research directions where, with longitudinal data and health and water microbial quality data, we will be able to conduct a comprehensive analysis as suggested by R2, including the links between risk substitution pathways, water quality and diarrheal disease.

Reviewer 3 Report

The manuscript concerns the important issue of the considering alternate pathways of drinking-water contamination: evidence of risk substitution from arsenic mitigation programs in rural Bangladesh. Authors highlighted the need for interventions to address risk substitution, particularly the increased use of contaminated surface water when access to drinking water is reduced. As the increasing ownership and decreasing distance to deep tubewells, although crucial, is not sufficient to achieve equity in safe drinking-water access across rural Bangladesh.

The following remarks should be referred to.

How in practice the results of the presented work can be used? This should be discussed in the point concerning discussion of the results.

How generalizable are the findings? Can they be applied to other areas? How dependent are they to specific characteristics of the region under examination. The human right to water and the quantities that manage it as a supply of "water of life" should have a presence in this text.

The novelty of the paper should be explicitly highlighted. Deepen the discussion of results comparing with other case studies that address the same theme. Are there concrete steps that can be recommended?

Author Response

Response to Reviewer 3 (R3)

We appreciate R3’s comments and recommendations, which motivated revisions to broaden the interest and impact of our work.

  1. How in practice the results of the presented work can be used?

R3 suggested a deeper discussion about the significance of our results and their translation to practice. This prompted us to expand our discussion section and devote a paragraph in section 4.2 (lines 291-316). We make two concrete points about how our results can be translated into practice. First, we based on our results, talk about the limitations of using ‘distance’ as a sole determinant for increasing equitable safe water provision. The problem of elite capture has been documented in rural Bangladesh where households with political influence are able to divert public funds to build tubewells in their compound and deter nearby households from using the deep tubewell extensively. Hence, we propose the continued involvement of governmental and non-governmental agencies to protect vulnerable households and provide equitable safe drinking water provision. Second, we suggest the development of user-friendly Water Safety Plans, that are lacking in many countries including Bangladesh, to aid households to practice safe drinking-water storage container practices. This suggestion is driven by our results that show that even if households procure water from deep tubewells owned by them, they are still at higher risk for surface water contamination in their drinking-water storage containers. Both of these points are concrete suggestions for policy makers.

  1. How generalizable are the findings? Can they be applied to other areas? How dependent are they to specific characteristics of the region under examination.

We agree with R3’s suggestion about discussing the generalizability of our results and have taken the following steps. First, we have added background information in section 2.2 (lines 114-125) about arsenic levels and tubewell composition in our study area so that users can keep the specific characteristic of our study area into account while evaluating the significance of our results. Second, we discuss the generalizability of our results in the discussion section 4.2 (lines 350-372). We first compare the context of our study area with the rest of southern Bangladesh where high arsenic concentrations in shallow aquifers and low groundwater depth make deep tubewells a suitable arsenic mitigation option. Then we broaden the discussion about sustainability of deep tubewells, and explicitly highlight concerns especially in areas with low arsenic concentrations, where risk of contamination or drying up of deep aquifers may be higher. Hence, our results would depend on the feasibility of deep tubewell installations as an arsenic mitigation strategy. In areas where deep tubewells are already used or are feasible to use as a safe drinking-water option our findings and their significance will be relevant.

  1. The novelty of the paper should be explicitly highlighted. Deepen the discussion of results comparing with other case studies that address the same theme. Are there concrete steps that can be recommended?

We agree with R3’s suggestion in explicitly highlighting the novelty and relevance of our paper and deepening the discussion of the results in comparison with other similar studies. We have expanded the discussion section substantially and have devoted a paragraph (section 4.2 lines 270-290) to the discussion. We mention that the findings add an important novel dimension to the evidence related to the idea of risk substitution that arsenic mitigation programs in Bangladesh and other affected countries may result in increases in health risks due to waterborne pathogens. Then we synthesize evidence on risk substitution from 6 different prominent studies. We emphasize that our study is different as it focuses on the actual pathways of risk substitution as compared to previous studies. Then we give a concrete example of why that might be important; even if diarrheal disease outcomes are similar among deep and shallow tubewell users, it is possible that the pathways attributed to diarrheal disease might be different. This can have significant implications for future WaSH interventions. Finally, in line with R3’s first suggestion about concrete steps, we write a paragraph detailing concrete steps for policy makers(section 4.2 (lines 291-316)).

Round 2

Reviewer 1 Report

The authors have taken an already very good manuscript and improved it far beyond that which was suggested by the reviewers.  Hopefully the authors' work will be noted by relevant policymakers (there's actually a chance of this since they are not writing about the U.S.!), and hopefully the authors will continue on this valuable trajectory to uncover the most important factor(s) associated with Bangladeshis' ability to access safe, affordable drinking water. 

Reviewer 3 Report

 Accept in present form.